# The Effect of Chronic Dietary Protein Manipulation on Amino Acids’ Profile and Position Sense in the Elderly Suffering from Type 2 Diabetes Mellitus

**DOI:** 10.3390/jfmk9020062

**Published:** 2024-04-04

**Authors:** Dionysia Argyropoulou, Tzortzis Nomikos, Gerasimos Terzis, Myrto Karakosta, George Aphamis, Nickos D. Geladas, Vassilis Paschalis

**Affiliations:** 1School of Physical Education and Sport Science, National and Kapodistrian University of Athens, 17237 Athens, Greece; argdiona@yahoo.gr (D.A.); gterzis@phed.uoa.gr (G.T.); ngeladas@phed.uoa.gr (N.D.G.); 2Department of Nutrition and Dietetics, Harokopio University, 17676 Athens, Greece; tnomikos@hua.gr (T.N.); ds220016@hua.gr (M.K.); 3Department of Life Sciences, University of Nicosia, Nicosia 1700, Cyprus; aphamis.g@unic.ac.cy

**Keywords:** elderly, protein intake, sarcopenia, type 2 diabetes mellitus

## Abstract

Dietary protein with adequate essential amino acids effectively stimulates protein synthesis and improves muscle mass. Musculoskeletal disorders in lower or upper limbs are not uncommon among patients with type II diabetes mellitus (T2DM). Therefore, this study primarily examines the effects of chronic dietary protein manipulation on amino acids’ profile and position sense in the elderly suffering from T2DM. A total of 26 individuals suffering from non-insulin-dependent T2DM (age > 55 years old) participated in a 12 week nutritional intervention. The subjects were randomly assigned and the control group received 0.8–1.0 g protein/kg/day, while the intervention group received 1.2–1.5 g protein/kg/day. Lean body mass, muscle strength, and position sense were assessed at baseline, as well as at the 6th and 12th week of the intervention. Only in the intervention group, the essential amino acids intake met the current nutritional recommendations (*p* < 0.05), while, by the 12th week, only the intervention group showed significant improvement in the muscle strength of knee (*p* < 0.05) and shoulder (*p* < 0.05) extension. On the contrary, in the control group, a significant decline in appendicular lean mass (*p* < 0.05) was observed by the 12th week. Position sense at the knee joint revealed a tendency for improvement in the intervention group by the 12th week (main effect of time *p* = 0.072). In the present investigation, it was revealed that the higher protein intake in the intervention group seemed to have positive effects on muscle strength and nearly positive effects on position sense.

## 1. Introduction

Type 2 diabetes mellitus (T2DM) is a common illness among the elderly [1] and is coined as a metabolic disorder characterized by ineffective insulin secretion or defective insulin action, or both [2]. However, low insulin levels may lead to insufficient protein synthesis and increased protein degradation, which, in turn, leads to muscle loss and maybe to sarcopenia [3,4].

Among the actions that should be followed for reducing the deleterious effects of T2DM is a diet rich in protein [5]. Indeed, low-carbohydrate/high-protein diets are recommended for patients suffering from T2DM, based on the rationale that blood glucose would decrease as a result of lower carbohydrate ingestion [6,7]. Ketogenic diets, that is, diets high in fat and protein intake, but low in carbohydrates, have also been reported to modulate the microglial inflammatory response, which, in turn, contributes towards neuroprotection and improves metabolic health [8]. Moreover, it is interesting that protein can improve insulin response without increasing plasma glucose concentrations [6,9]. Amino acids derived from proteins, in combination with glucose, stimulate insulin secretion, leading to protein synthesis and an increase in muscle mass [10]. Note, however, that, unless amino acids intake is balanced, adequate protein supplementation is not necessarily ingested effectively [11]. That is, not all amino acids will definitely contribute to protein synthesis. Nonetheless, it is clear that the anabolic effect of insulin on muscle mass and the adequate amino acids intake (meals with high quality proteins) could be an effective countermeasure for muscle loss [4,12].

It can be hypothesized that a diet rich in protein may provide a wide range of amino acids, including the essential ones, which, in turn, may improve both the strength and physical performance of the elderly and those with insulin sensitivity [13]. Four amino acids (i.e., leucine, isoleucine, alanine, and arginine) were found to be particularly important for stimulating β-cell electrical activity, essential for insulin secretion [14].

The existing guidelines for dietary proteins suggest 0.8 g of protein per kilogram of body mass per day (g/kg/day), independent of sex or age [14]. However, in the PROT-AGE project, it was proposed that an average daily protein intake of 1.2–1.5 g/kg/day (or approximately 25–30 g/meal) should be consumed by older adults suffering from acute or chronic diseases, depending on the characteristics of the disease (e.g., the severity of the disease and the impact of the disease on the patient’s diet) [15]. This nutritional approach could have positive effects in patients suffering with T2DM, since poor muscle strength is often observed in these patients [16,17], since skeletal muscle mass is heavily affected by the defective insulin mechanism [18,19,20].

Musculoskeletal disorders in lower or upper limbs are not uncommon among patients with T2DM [21,22,23]. Inevitably, knee and shoulder joints have been repeatedly reported to be affected by T2DM, resulting in deteriorated muscle strength and reduced mobility of the individuals [21,22,23]. An important parameter that can be affected by muscle strength reduction is the proprioception, which is characterized as the sixth sense, according to Sir Charles, and describes the ability of the human body for orientation, position, and motion sensing [24]. Indeed, low proprioception functions/thresholds and specifically inadequate position sense have been linked with lower body balance, perturbations in daily activities, and increased risks for musculoskeletal injuries—injury recurrences [24]. It is a common experience that we feel unsteady on our legs and have difficulty in performing fine movements with our hands (like writing or drawing) after participating in physical activities that cause fatigue. Position sense is one of the main parameters for the evaluation of proprioception. Disturbed position sense has been found after exercise performed either with the legs [24,25,26] or with the arms [27,28,29,30]. Interestingly, it has been observed that baseline position sense in overweight individuals was worse than their lean counterparts [27], which, in turn, may lead to disturbed motor performance.

It was suggested that a chronic unhealthy condition (e.g., increased body composition or T2DM) may lead to the development of reduced position sense, increasing the number of falls and serious injuries during everyday activities. Indeed, few recent studies have reported that T2DM patients exhibit impaired hip joint or knee joint position sense [31,32] and, while sensorimotor training is effective to improve proprioception [33], no study has been found to take protein or any other diet into consideration.

Dietary protein stimulates protein synthesis and improves muscle mass. However, dietary protein might not be ingested effectively and, therefore, one of the aims of the present investigation was to create the profile of the essential amino acids of the two diets (differing only in the protein intake) in order to assess their quality. Moreover, since T2DM patients often exhibit musculoskeletal disorders in the lower or upper limbs, and since dietary protein improves muscle mass, this study also aimed to assess the effect of the two diets on the muscle strength and position sense of the knee and shoulder joints. It has been hypothesized that essential amino acid nutritional recommendations will be reached with both diets, since either diet reaches the recommended daily nutritional protein intake [34]. However, since muscle mass is positively affected by protein intake and due to the fact that T2DM patients tend to have impaired joints, it was hypothesized that muscle strength and, consequently, position sense would be improved only in the group with the higher protein intake, while in the group with the lower protein intake, the position sense and the muscle strength would probably remain stable.

## 2. Materials and Methods

### 2.1. Participant Description and Eligibility Criteria

Participants diagnosed with T2DM (non-insulin-dependent) during the last 10 years, aged between 50 and 75 years old were recruited (Table 1). All female participants were post-menopausal. Exclusion criteria included (a) receiving dietary supplements or following another dietary intervention within one month prior to or during the study, (b) having nephropathy or cancer, (c) experiencing cognitive impairments, (d) having neurological disorders, (e) undergoing previous surgery that would deteriorate their motion during the assessments, and (g) having a body mass index (BMI) > 40. Prior to inclusion, all participants provided written informed consent. The study protocol was approved by the National and Kapodistrian University of Athens ethics committee (Protocol Record 1288/03-07-2021) and was registered at clinicaltrials.gov as NCT05457088.

### 2.2. Study Design

Participants were randomly assigned into two parallel groups, namely the control and the intervention group. Over a period of 12 weeks, participants followed one of two dietary interventions, differing in the amount of dietary protein, while maintaining a comparable total daily energy intake. In the control group (CG), participants were given a diet plan with the current recommended protein intake (0.8–1.0/g/kg/day), whereas in the intervention group (IG), participants followed a diet plan with a higher protein intake, ranging from 1.2 to 1.5 g/kg/day. Before the beginning of the study, participants underwent measurements of their anthropometric characteristics, appendicular lean mass, and complete blood count, as well as their nutritional habits. Additionally, their muscle strength and position sense were evaluated. At week 0, baseline measurements were performed and participants were provided with their respective dietary plan based on their assigned group. At the end of the intervention period (i.e., week 12), all measurements were repeated. To ensure adherence to the diet plan, follow-up measurements were conducted at week 6, excluding body composition measurements. A physical activity questionnaire was completed by participants at weeks 0 and 12, in order to ensure that the changes were a product of our intervention and not due to changes in physical activity [35].

### 2.3. Dietary Assessment

Assessment of nutritional habits included the analysis of a 3-day dietary recall at weeks 0, 6, and 12 for the macronutrient analysis of diet (i.e., protein, carbohydrates, and lipids). Through diet analysis, we also checked whether the participants followed the instructed diet. The 3-day recall data were collected [36] and for the nutrition analysis, we used the FoodData database (U.S Department of Agriculture) [37].

### 2.4. Anthropometric Characteristics and Appendicular Lean Mass Assessment

A weight scale (Seca 700, Hamburg, Germany) was used to measure both body mass and height. The body mass index (BMI) was calculated as the ratio between body mass (kg) and the square of height (m^2^). The waist to hip ratio is used in the literature as a reliable marker for risks related to fatal diseases [38,39]. A flexible and inextensible anthropometric tape (Seca 201, Hamburg, Germany) was used to measure waist and hip circumferences. All the above measurements were assessed at weeks 0, 6, and 12.

Appendicular lean mass was determined using dual energy X-ray absorptiometry (DXA, Genaral Electric, Luna prodigy, CA, USA). A DXA scan is the most reliable way to assess the body composition of the participants [40]. Measurements of appendicular lean mass were taken at weeks 0 and 12. At the 6th week of intervention, an appendicular lean mass measurement was omitted to minimize participants’ exposure to radiation. All measurements were carried out by the same trained technician and the equipment was calibrated daily, according to the manufacturer’s specifications.

### 2.5. Hematological Measurements

After a 12 h overnight fast, blood samples were obtained for hematological tests between 8.30 and 10.30 the following morning at weeks 0, 6, and 12. In order to ensure compliance with fasting, reminders were sent on the previous day. Complete blood count was determined in EDTA anticoagulated whole blood with a Mindray BC-3000 hematology analyzer (Mindray, Shenzhen, China). The blood sample was evaluated by experienced nurses who were informed of the methodology. The white blood count (WBC) and platelet count (PLT) are associated with inflammation and to several cardiovascular disease risk factors and diabetes [41]. The consistent elevation of HbA1c can be associated with functional and structural changes in the hemoglobin molecule, cytoplasmic viscosity, and osmotic disturbances within red blood cells. These changes may be reflected in the red blood cell analytical parameters such as hematocrit (HCT), red blood cells count (RBC), and hemoglobin concentration (HGB), making diabetic patients prone to develop mild anemia [42].

### 2.6. Muscle Strength Assessment

The participants had their muscle strength measured at weeks 0, 6, and 12 using a portable force evaluation and testing dynamometer (microFET2, Hoggan, UT, USA). The procedures were fully explained before assessment, followed by a familiarization attempt.

Knee extension—participants were sitting with the hip and knee flexed at 90°. The subject was asked to extend their knee. The dynamometer was placed proximal to maleoli on the front side of the leg.

Shoulder extension—participants were in a supine position with the shoulder flexed at 90°. The subject was asked to push the shoulder downwards for the extension movement. The dynamometer was placed proximal to the wrist on the back side of the arm. The intraclass correlation coefficients for knee extension and shoulder extension were 0.97 and 0.99, respectively.

### 2.7. Position Sense Assessment

Positions sense was evaluated in both the knee and shoulder joints, following protocols from previous studies of a similar nature, albeit utilizing different machinery [25,26,27,43,44]. In the present investigation, for the assessment of the position sense of the knee joint, the subjects were seated on a chair with their back straight and hands crossed in front of their chest. During the assessment of the position sense of the shoulder joint, the subjects stood sideways against the wall in their anatomical position. The target position was marked on the wall by a thin highlighted line, with the exact spot identified individually for each participant. Specifically, the target position for the knee joint was determined by the point at which the participant touched the wall with their tiptoe, with the knee joint flexed at 90°, while for shoulder joint abduction, it was the point at which the participants touched the wall with their fingertips, with the shoulder abducted at 90°. The intraclass correlation coefficients for knee proprioception and shoulder proprioception were 0.91 and 0.94, respectively.

During the assessment, the movement for the placement of the limb was performed by the subject, but the position was fixed by the investigator. Specifically, the investigator positioned the lower or the upper limb on the targeted position for 10 s and returned the limb to the starting position. Subsequently, participants were asked to recall the reference position by actively moving their limb to the target angle and when they were satisfied with the selected position held it for about 2 s. The investigator, using a flexible and inextensible anthropometric tape (Seca 201, Hamburg, Germany), measured the distance between the position chosen by the subjects and the target position, which represents an index of the magnitude of matching error.

The subjects completed three efforts for both knee and shoulder joints, with the mean of the two closest to the reference position being recorded and used for the statistical analysis. All assessments were performed without visual feedback (e.g., blindfolded). All the position sense assessments were performed on the dominant limbs, with all procedures performed by the same person.

### 2.8. Statistical Analysis

Normality was tested with the Kolmogorov–Smirnov criterion. Normally distributed continuous variables were presented as means ± standard deviation. Comparisons of the baseline characteristics of our population were based on ANOVA analysis for normally distributed variables; post hoc analysis was carried out with the Bonferroni correction. For the analyses of group, time, and group by time effect, a repeated measures ANOVA was performed for normally distributed continuous variables. All reported *p*-values were based on two-sided tests and compared to a significance level of 5%. Bonferroni correction was used for pairwise comparisons among the two study groups. The intraclass correlation coefficient was calculated through a random-effect two-way analysis of variance (ANOVA) model. All analyses were conducted using the IBM SPSS Statistics software, version 25.0 (IBM Corp., Armonk, NY, USA).

## 3. Results

### 3.1. Dietary Assessment

Participants of both groups successfully followed the scheduled dietary plans (Table 2). In particular, the control group received dietary protein at 0.8–1 g/kg/day, while the intervention group received dietary protein at 1.2–1.5 g/kg/day. Protein intake per body mass was significantly different between the two groups after week 12 of intervention (*p* < 0.05). Specifically, the relative daily protein intake between baseline and week 12 was not significant in the control group (*p* = 0.13), while it was significant in the intervention group (*p* < 0.05) (by study design). As was expected, a significant group by time interaction and a significant difference between groups was observed, in accordance with study design.

Regarding the composition of protein intake into amino acids (Table 2), the control group failed at most recommendations, with the exception of isoleucine, valine, and a methionine + cysteine combo. No significant results were obtained over time. In contrast, the intervention group succeeded to meet all of the amino acids’ intake recommendations. In particular, the intervention group significantly increased in tryptophan (*p* < 0.01), threonine (*p* < 0.01), isoleucine (*p* < 0.01), leucine (*p* < 0.01), lysine (*p* = 0.01), valine (*p* < 0.01), histidine (*p* = 0.01), methionine + cysteine combo (*p* < 0.01), and phenylalanine + tyrosine combo (*p* < 0.01) over time. Significant differences were observed between the groups across all assessments of amino acid intake.

### 3.2. Anthropometric Characteristics and Appendicular Lean Mass Assessment

Nutrition intervention did not induce any significant changes in weight, BMI, and waist to hip ratio through weeks 0, 6, and 12, compared to the baseline for both groups (*p* > 0.05) (Table 3). On the contrary, the circumference of the waist significantly reduced over time only in the intervention group (*p* < 0.05).

Regarding appendicular lean mass (Table 3), the measurements revealed that the control group experienced a significant loss over time (*p* < 0.05), whereas the intervention group successfully maintained its appendicular lean mass. Additionally, a significant group by time interaction was observed (*p* < 0.05).

### 3.3. Hematological Measurements

The blood sample analysis did not show any significant differences in participants between groups across all weeks of intervention (Table 4).

### 3.4. Muscle Strength Assessment

A significant group by time interaction was observed in knee extension (*p* < 0.05) and shoulder extension (*p* < 0.05) assessments, while there was a significant effect of the group in the strength assessment of shoulder extension (*p* < 0.05).

Regarding the control group, muscle strength performance significantly deteriorated over time in both knee extension (*p* < 0.05) and shoulder extension (*p* < 0.05) assessments (Table 5). On the contrary, in the intervention group, muscle strength significantly improved in both knee extension (*p* < 0.05) and shoulder extension (*p* < 0.05) assessments (Table 5).

### 3.5. Position Sense Assessment

A protein-rich diet appeared to have a tendency for positive effects on position sense, particularly evident in the knee joint. Specifically, in the experimental group, there was a tendency for the positive effect of time on knee joint position sense (*p* = 0.072). There was no main effect of group or in the interaction between time and group. Conversely, there were no significant main effects of time or group, while there were no integrations between time and group in the assessment of shoulder position sense (Figure 1B).

## 4. Discussion

In the present investigation, it was hypothesized that both diets (0.8 g protein/kg/day and 1.5 g protein/kg/day) would meet essential amino acid nutritional recommendations and that improvements in position sense and muscle strength would be improved only in the group with the higher protein intake. Contrary to our hypothesis, the current nutritional recommendations were only met in the intervention group, regarding amino acid intake. Interestingly, by the 12th week of intervention, a significant decline in lean body mass was observed in the control group, while the experimental group demonstrated a significant improvement in muscle strength and a marginally not significant improvement in position sense of the knee joint.

A protein-rich diet can be beneficial for patients with T2DM by aiding in glucose regulation and ameliorating muscle mass loss for the elderly [45]. Current recommendations suggest that adults should consume 0.8–1.0 g/kg/day protein; nevertheless, there is a growing trend advocating for higher protein intake levels of 1.2–1.5 g/kg/day, particularly for older adults and especially in the case of chronic diseases [15]. As was the case in the present investigation, protein ingestion of 0.8–1 g/kg/day was insufficient to meet the recommended intake values of essential amino acids (except isoleucine, valine, and a methionine + cysteine combo) [34]. On the contrary, a protein intake of 1.2–1.5 g/kg/day was found to significantly improve amino acids’ profile, leading to an improved quality of protein intake. Moreover, only in the protein-rich diet did the amino acid leucine meet adequate intake requirements. This finding is noteworthy as leucine is associated with an increased anabolic response [46]. It is important to create diets that meet the amino acids’ requirements, since a lack of several essential amino acids leads to an increased risk of pre-diabetes [47].

It is well established that an adequate protein intake is essential in maintaining a balance between protein synthesis and breakdown [48,49]. Indeed, in the present study, the intervention group exhibit a preservation of appendicular lean body mass and a significant reduction in waist circumference over the period of the 12 weeks. In contrast, the control group exhibited a significant reduction in appendicular lean body mass, indicating a negative balance between protein synthesis and breakdown. Our data are not in line with studies which found a positive relationship between protein intake and muscle synthesis [50,51]. This discrepancy could be attributed to the bidirectional relationship of T2DM with sarcopenia, meaning that the T2DM amplifies the muscle mass loss [3], a condition which may enhance muscle breakdown.

In the intervention group, the observed preservation of muscle mass and the increased protein intake was accompanied by improvements in muscle strength of the knee and shoulder extensors. Conversely, the control group exhibited a decline in muscle strength during assessments in the same muscle groups. Our findings indicate that a protein intake of 0.8–1.0 g protein/kg/day is simply not enough for the elderly suffering from T2DM, in order to maintain their muscle strength. However, doubling the recommended protein intake had beneficial effects on muscle strength, as was the case in studies of a similar nature [52,53,54].

Proprioception is considered as one of the subsystems of the somatosensory system. In the present investigation, it was found that individuals suffering from T2DM following a dietary schedule with increased protein intake tended to exhibit an improved position sense in the knee joint, whereas no such effect was observed in the control group. These results provide direct evidence suggesting that the level of protein intake may marginally enhance position sense in the lower limbs. The importance of the present findings relies on the fact that adequate position sense is required for capable and delicate human movement. Disturbances in proprioception can increase the risk of injuries during everyday life activities due to reduced motor control. Indeed, it is a common experience that there is difficulty in performing common movements after participating in physical activities that cause fatigue, which, in turn, may increase the risk of injuries [25,26,27,55,56,57]. To the best of our knowledge, no studies have been found to examine the effect of protein on position sense.

Overall, this study revealed that for patients suffering from T2DM, a diet with a protein intake of 1.2–1.5 g/kg/day is more effective than the current recommendations (0.8–1.0 g/kg/day) to meet essential amino acids’ requirements and stimulate protein synthesis. The increase in protein synthesis explains the significant improvements in muscle strength for the group with the higher protein intake, in contrast to the group with the current recommended intake. A limitation of the present study was that the possible changes in the central peripherical neural system, and structures and functions of the muscular system, which seems to affect position sense, were not evaluated. Another limitation is that the recording of dietary recalls relied on the memory and perception of each individual. More studies with a larger sample size are needed to explore the effects of T2DM, which may contribute to sarcopenia, on position sense. Additionally, while this investigation noted a tendency towards improved lower leg position sense, it is worth noting that upper extremity musculoskeletal disorders are a common and understudied problem in patients with diabetes mellitus [58]. It is essential that future nutritional interventions include a method to assess the quality of the dietary plan. Finally, it is known that exercise can benefit muscle mass and, thus, a combination of protein diet with exercise would probably provide better results in both muscle strength and position sense.

## 5. Conclusions

A protein intake in an amount of 1.2–1.5 g/kg/day (intervention group) can significantly improve the quality of food and also substantially improve muscle strength performance. In contrast, a protein intake of 0.8–1.0 g/kg/day (control group) was insufficient to meet current recommendations for essential amino acids’ intake. Furthermore, this lower protein intake failed to mitigate the loss of muscle mass associated with aging and resulted in decreased muscle strength performance.

It is evident that including physical performance measures in studies involving T2DM patients is crucial for assessing the risk of disability and evaluating the effectiveness of preventive strategies [59]. Additionally, studies, in this context, could re-evaluate current recommendations for daily protein intake.

## Figures and Tables

**Figure 1 jfmk-09-00062-f001:**
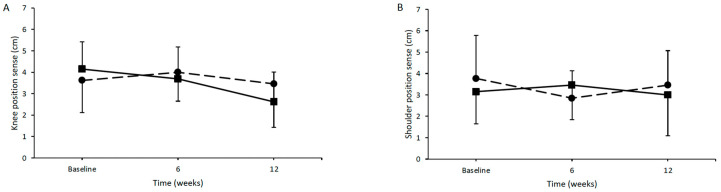
(**A**) Average position sense of knee joints at baseline, after 6 weeks, and after 12 weeks of intervention (control group: dashed line, intervention group: continuous line). (**B**) Average position sense of shoulder joints at baseline, after 6 weeks, and after 12 weeks of intervention (control group: dashed line, intervention group: continuous line).

**Table 1 jfmk-09-00062-t001:** Baseline anthropometrics of participants (n = 26).

	Control Group (Men = 6, Women = 7)	Intervention Group (Men = 7, Women = 6)
Age (years)	59.31 ± 6.82	60.62 ± 6.84
Height (cm)	166.15 ± 11.53	168.85 ± 9.38
Weight (kg)	92.27 ± 24.26	80.88 ± 13.91
Body mass index	33.16 ± 7.48	28.34 ± 4.29

**Table 2 jfmk-09-00062-t002:** Energy consumption, protein intake, and amino acid intake between groups at baseline, after 6 weeks, and after 12 weeks of the intervention. Current recommendations for amino acids’ profile are also included for comparison [34].

	Group	Baseline	6 Weeks	12 Weeks	*p* _time_	*p* _group_	*p* _time*group_	R
Energy (kcal)	CG	2052 ± 616	2137 ± 623	2102 ± 569	0.40	0.60	0.89	-
	IG	1947 ± 181	2032 ± 246	2027 ± 257	0.25			
Protein (g/kg/day)	CG	0.80 ± 0.22	0.85 ± 0.20	0.94 ± 0.16	0.13	<0.05	<0.05	-
	IG	0.81 ± 0.17	1.21 ± 0.18 ^a^	1.42 ± 0.12 ^bc^	<0.05			
Tryptophan (mg/kg/day)	CG	5.75 ± 2.33	6.37 ± 2.98	7.34 ± 3.01	0.27	<0.05	0.16	8
	IG	6.06 ± 2.56	8.25 ± 2.45 ^a^	9.87 ± 2.99 ^b^	<0.05			
Threonine (mg/kg/day)	CG	20.24 ± 8.80	21.50 ± 10.32	25.55 ± 10.91	0.30	<0.05	0.18	28
	IG	21.03 ± 9.98	28.37 ± 9.53 ^a^	34.16 ± 11.02 ^b^	<0.05			
Isoleucine (mg/kg/day)	CG	22.28 ± 10.20	24.10 ± 11.82	28.99 ± 13.80	0.28	<0.05	0.21	23
	IG	22.84 ± 11.04	31.11 ± 10.47 ^a^	38.49 ± 12.92 ^b^	<0.05			
Leucine (mg/kg/day)	CG	37.19 ± 15.48	40.28 ± 19	47.19 ± 20.26	0.30	<0.01	0.23	49
	IG	40.4 ± 19.13	53.98 ± 18.69	63.74 ± 20.35 ^b^	<0.05			
Lysine (mg/kg/day)	CG	34.09 ± 17.63	35.12 ± 19.56	44.24 ± 21.75	0.31	<0.05	0.29	48
	IG	36.07 ± 19.12	46.71 ± 19.54	60.51 ± 21.12 ^bc^	<0.05			
Valine (mg/kg/day)	CG	24.71 ± 9.93	26.80 ± 12.66	32.02 ± 14.19	0.25	<0.05	0.33	32
	IG	27.21 ± 13.38	35.36 ± 12.46	42.16 ± 14.47 ^b^	<0.05			
Histidine (mg/kg/day)	CG	14.12 ± 7.04	14.28 ± 6.64	16.79 ± 7.47	0.51	<0.05	0.11	18
	IG	14.01 ± 6.22	19.42 ± 7.62	23.5 ± 6.97 ^b^	<0.05			
Methionine + cysteine (mg/kg/day)	CG	18.60 ± 8.39	19.95 ± 10.19	23.94 ± 10.82	0.27	<0.05	0.21	23
	IG	19.40 ± 9.85	25.96 ± 9.36 ^a^	31.84 ± 10.77 ^b^	<0.05			
Phenylananine + tyrosine (mg/kg/day)	CG	37.06 ± 14.98	39.90 ± 18.78	47.09 ± 20.40	0.27	<0.05	0.23	48
	IG	39.43 ± 18.67	52.53 ± 18.10 ^a^	61.92 ± 20.53 ^b^	<0.05			

CG: control group; IG: intervention group; R: recommendation. ^a^: *p* < 0.05 compared 6 weeks to baseline. ^b^: *p* < 0.05 compared 12 weeks to baseline. ^c^: *p* < 0.05 compared 6 weeks to 12 weeks.

**Table 3 jfmk-09-00062-t003:** Anthropometric characteristics between groups at baseline, after 6 weeks, and after 12 weeks of the intervention.

	Group	Baseline	6 Weeks	12 Weeks	*p* _time_	*p* _group_	*p* _time*group_
Weight (kg)	CG	92.27 ± 24.26	91.91 ± 23.64	91.25 ± 21.97	0.56	0.17	0.66
	IG	80.88 ± 13.91	81 ± 14.04	80.68 ± 14.06	0.39		
Body mass index	CG	33.16 ± 7.48	33.02 ± 7.06	32.77 ± 6.12	0.54	0.06	0.61
	IG	28.34 ± 4.29	28.40 ± 4.47	28.28 ± 4.47	0.40		
Waist circumference (cm)	CG	103.77 ± 17.81	103.23 ± 18.14	102.5 ± 16.07	0.50	0.33	0.67
	IG	97.81 ± 14.18	96.38 ± 14.13 ^a^	96.23 ± 13.98 ^b^	<0.05		
Hip circumference (cm)	CG	112.27 ± 13.89	111.69 ± 14.37	110.92 ± 13.29	0.30	0.20	0.98
	IG	106.15 ± 8.79	105.54 ± 8.98	104.73 ± 8.19	0.07		
Waist/hip ratio	CG	0.92 ± 0.09	0.92 ± 0.09	0.92 ± 0.09	0.90	0.85	0.61
	IG	0.92 ± 0.09	0.91 ± 0.08 ^a^	0.92 ± 0.09	0.25		
Appendicular lean mass (kg)	CG	22.78 ± 6.35	-	20.95 ± 6.38 ^b^	<0.05	0.51	<0.05
	IG	20.08 ± 4.66	-	20.57 ± 5.04	0.29		

CG: control group; IG: intervention group. ^a^: *p* < 0.05 compared 6 weeks to baseline. ^b^: *p* < 0.05 compared 12 weeks to baseline.

**Table 4 jfmk-09-00062-t004:** Hematological measurements between groups at baseline, after 6 weeks, and after 12 weeks of the intervention.

	Group	Baseline	6 Weeks	12 Weeks	*p* _time_	*p* _group_	*p* _time*group_
WBC (10^3^/uL)	CG	6.90 ± 1.94	7.58 ± 1.90	7.41 ± 1.74	0.50	0.140	0.50
	IG	5.99 ± 1.88	6.78 ± 2.65	6.25 ± 2.17	0.14		
HGB (g/dL)	CG	14.61 ± 2.88	16.80 ± 5.92	14.64 ± 1.51	0.26	0.38	0.86
	IG	14.57 ± 2.03	14.24 ± 1.87	15.21 ± 4.67	0.53		
RBC (10⁶/uL)	CG	4.87 ± 0.74	5.61 ± 1.51	5.19 ± 0.61	0.24	0.25	0.88
	IG	5.09 ± 0.52	6.28 ± 3.53	5.34 ± 1.21	0.45		
HCT (%)	CG	44.54 ± 8.48	51.37 ± 16.6	44.91 ± 3.59	0.25	0.37	0.17
	IG	44.51 ± 5.24	43.46 ± 4.57	46.61 ± 13.14	0.49		
PLT (10^3^/uL)	CG	262.00 ± 97.88	274.00 ± 140.87	281.43 ± 79.36	0.58	0.67	0.29
	IG	240.00 ± 56.57	257.50 ± 38.43	227.13 ± 58.50	0.23		

CG: control group; HCT: hematocrit; HGB: hemoglobin concentration; IG: intervention group; PLT: platelet count; RBC: red blood cells count; WBC: white blood cells count.

**Table 5 jfmk-09-00062-t005:** Muscle strength between groups at baseline, after 6 weeks, and after 12 weeks of the intervention.

	Group	Baseline	6 Weeks	12 Weeks	*p* _time_	*p* _group_	*p* _time*group_
Knee extension (kg)	CG	27.60 ± 8.30	26.00 ± 7.40	25.10 ± 8.50 ^b^	<0.05	0.613	<0.05
	IG	26.10 ± 6.20	27.60 ± 6.60	29.50 ± 7.00 ^bc^	<0.05		
Shoulder extension (kg)	CG	16.80 ± 7.70	16.70 ± 7.80	15.70 ± 7.10	<0.05	<0.05	<0.05
	IG	14.00 ± 4.70	14.40 ± 4.80	15.60 ± 5.10 ^bc^	<0.05		

CG: control group; IG: intervention group. ^b^: *p* < 0.05 compared 12 weeks to baseline. ^c^: *p* < 0.05 compared 6 weeks to 12 weeks.

## Data Availability

The data presented in this study are available on request from the corresponding author (Vassilis Paschalis; email address: vpaschalis@phed.uoa.gr).

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
