# Peer review of "The Effect of Chronic Dietary Protein Manipulation on Amino Acids’ Profile and Position Sense in the Elderly Suffering from Type 2 Diabetes Mellitus"

_jfmk, 2024, doi:10.3390/jfmk9020062_

Round 1

Reviewer 1 Report

Comments and Suggestions for Authors

The double aim of the present investigation was to create the profile of the essential amino acids for two diets with different protein intake and 72 assess the effect of the diets on position sense of the knee and the shoulder joints. It has been hypothesized that essential amino-acid nutritional recommendations will be reached with both diets, since either reach the daily nutritional recommendation protein intake [25]. However, since muscle mass is positively affected by proteins intake, it was hypothesized that muscle strength and consequently position sense would be improved only in the group with the higher protein intake. 

The manuscript is well structured and deals with a topic of great relevance and potential interest for the scientific community.

I only have a few minor suggestions for authors.

In particular, regarding the introductive paragraph, I believe it is appropriate to remodulate the purpose of the study. The authors should formulate the purpose of the study more clearly and also provide a clear research hypothesis.

Regarding the introduction, to improve, the authors could also take into consideration the following study:

-       Polito et al., The Ketogenic Diet and Neuroinflammation: The Action of Beta-Hydroxybutyrate in a Microglial Cell LineInternational Journal of Molecular Sciences, 2023, 24(4), 3102

The tables need substantial modifications. In particular, it must be made legible. Furthermore, the authors should report a detailed but concise description of what is reported in the tables in the captions of the figures. Furthermore, it is necessary to specify all the acronyms used.

The figures are low resolution and therefore appear blurry.

The "Discussions" paragraph begins like this: The aim of the present investigation was to examine the effects of chronic dietary protein manipulation on amino acids profile and position sense in elderly suffering type II diabetes mellitus. 

In this section the authors should not restate the purpose of the study but begin by highlighting the main findings of the study. Subsequently the results must be discussed on the basis of what is present in the literature.

Future implications should also be clearly stated in the conclusions

Author Response

Reviewer 1

The double aim of the present investigation was to create the profile of the essential amino acids for two diets with different protein intake and assess the effect of the diets on position sense of the knee and the shoulder joints. It has been hypothesized that essential amino-acid nutritional recommendations will be reached with both diets, since either reach the daily nutritional recommendation protein intake [25]. However, since muscle mass is positively affected by proteins intake, it was hypothesized that muscle strength and consequently position sense would be improved only in the group with the higher protein intake. The manuscript is well structured and deals with a topic of great relevance and potential interest for the scientific community.

Response

We would like to thank the reviewer for his/her supportive comments about our manuscript and for the comments that helped us improve our work.

I only have a few minor suggestions for authors.

In particular, regarding the introductive paragraph, I believe it is appropriate to remodulate the purpose of the study. The authors should formulate the purpose of the study more clearly and also provide a clear research hypothesis.

Response

According to the reviewer’s comment, the purpose of the study along with the research hypothesis were remodulate and now the relevant text reads:

“Based on the above information, Dietary protein stimulates protein synthesis and improves muscle mass. However, dietary protein might not be ingested effectively, and therefore one of the aims of the present investigation was to create the profile of the essential amino acids of the two diets (differing only in the protein intake) in order to assess their quality. the double aim of the present investigation was to create the profile of the essential amino acids for two diets with different protein intake and assess the effect of the diets on position sense of the knee and the shoulder joints. Moreover, since T2DM patients often exhibit musculoskeletal disorders in lower or upper limbs and since dietary protein improves muscle mass, this study also aimed to assess the effect of the two diets on position sense of the knee and shoulder joints. It has been hypothesized that essential amino-acid nutritional recommendations will be reached with both diets, since either reach the daily nutritional recommendation protein intake [25]. However, since muscle mass is positively affected by proteins intake and due to the fact that T2DM patients tend to have impaired joints, it was hypothesized that muscle strength and consequently position sense would be improved only in the group with the higher protein intake, while in the group with the lower protein intake the position sense and the muscle strength would be probably remain stable.” (Introduction, par. 7)

Regarding the introduction, to improve, the authors could also take into consideration the following study:

-       Polito et al., The Ketogenic Diet and Neuroinflammation: The Action of Beta-Hydroxybutyrate in a Microglial Cell LineInternational Journal of Molecular Sciences, 2023, 24(4), 3102

Response:

The suggested study by the reviewer has been added in the introductory section to enforce the choice for high protein/low carbohydrates diet and now the relevant text reads:

“Ketogenic diets, that is, diets high in fat and protein intake but low in carbohydrates, have also been reported to modulate the microglial inflammatory response which in turn contributes in neuroprotection and improves metabolic health [8].” (Introduction, par. 2, lines 40-43).

The tables need substantial modifications. In particular, it must be made legible. Furthermore, the authors should report a detailed but concise description of what is reported in the tables in the captions of the figures. Furthermore, it is necessary to specify all the acronyms used.

Response

All tables in the revised manuscript were modified for been clearer to the reader. Specifically, the tables’ captions were changed for giving a more detailed description of each table while the acronyms were reduced and the remained ones are clearly explained. Moreover, we also modified the figure legend.

The figures are low resolution and therefore appear blurry.

Response

Figures of higher resolution are now included in the revised manuscript (Figure 1, panels A and B).

The "Discussions" paragraph begins like this: The aim of the present investigation was to examine the effects of chronic dietary protein manipulation on amino acids profile and position sense in elderly suffering type II diabetes mellitus. In this section the authors should not restate the purpose of the study but begin by highlighting the main findings of the study. Subsequently the results must be discussed on the basis of what is present in the literature.

Response

In the opening paragraph of the revised manuscript, the purpose of the study was omitted as suggested by the reviewer, and now only the hypothesis and the main findings are presented. Moreover, we added information of the present literature in order to discuss our results. The two first paragraphs of the Discussion section now read:

“In the present investigation, it was hypothesized that both diets (0.8 gram protein/kg/day and 1.5 gram protein/kg/day) would meet essential amino-acid nutritional recommendations, and that improvements in position sense and muscle strength would be improved only in the group with the higher protein intake. Contrary to our hypothesis, only in the intervention group the current nutritional recommendations were met regarding amino acid intake. Interestingly, by the 12th week of intervention a significant decline in lean body mass was observed in the control group, while the experimental group significant improvement in muscle strength and a marginally not significant improvement in position sense of the knee joint were observed.

A protein-rich diet can be beneficial for patients with T2DM by aiding in glucose regulation and ameliorating muscle mass loss for elderly [45]. Current recommendations suggest that protein adults should consume 0.8-1.0 gram/kg/day, nevertheless there is a growing trend advocating for higher protein intake levels of 1.2-1.5gram/kg/day, particularly for older adults and especially in the case of chronic diseases [15]. As it was the case in the present investigation, protein ingestion of 0.8-1 gram/kg/day was insufficient to meet the recommended intake values of essential amino-acids (except of isoleucine, valine and methionine + cysteine combo [34]. On the contrary, protein intake of 1.2-1.5gram/kg/day was found to significantly improve amino-acids’ profile, leading to improve quality of protein intake. Moreover, only in the protein-rich diet the amino acid leucine met the adequate intake. This finding is noteworthy as leucine is associated with an in-creased anabolic response [46]. It is important to create diets that meet the amino acids’ requirements, since lack of several essential amino acids leads to increased risk of pre-diabetes [47].”

Future implications should also be clearly stated in the conclusions

Response

As suggested by the reviewer, future implications were added in the revised manuscript:

“It is essential that future nutritional interventions include a method to assess the quality of the dietary plan. Finally, it is known that exercise can benefit muscle mass and, thus, a combination of protein diet with exercise would probably provide better results in both muscle strength and position sense.” (Discussion, par. 6, lines 1050-1053).

Reviewer 2 Report

Comments and Suggestions for Authors

Dear authors,

This study was conducted to effects of chronic dietary protein manipulation on amino acids profile and position sense in elderly suffering type 2 diabetes mellitus. Fundamentally, this study is well described and use adequate methods and analyses. Moreover, it presents good data on this very important topic. I believe that this is very important issue in field of sports medicine and nutrition section. However, it needs more revisions. Some suggestions are recommended for the authors’ consideration.

Abstract

(1) Please you should add background of this study (1-2 sentences) in Introduction section.

(2) Line 22: “Position sense at the knee joint marginally improvement only in the intervention group by the 12th week (main effect of time p=0.072).” You have to revise “improvement” term because the p-value 0.072 is not statistically significant.

(3) Line 16: “1.0gr protein/kg/d and the intervention group received 1.2-1.5gr” à1.0 gram protein/kg/day and the intervention group received 1.2-1.5 gram protein/kg/day”

Introduction

(1) The authors should more explain the concept of chronic dietary protein manipulation on amino acids profile and position sense. Besides, the research gap between chronic dietary protein manipulation on amino acids profile and position sense in elderly and the authors should strengthen the gap illustration according to the prior research (added 3-4 paragraphs).

(2) If it is possible, please revise from “gr protein/kg/d” to “gram protein/kg/day” in whole manuscript.

Materials and methods: Well-written

(1) Line 88: Please provide exact IRB institute. It is not proper “local university ethics committee”.

(2) In Table 1, Please revise “BMI” to “body mass index”.

(3) Line 119: Please revise “m2” to “m2” in whole manuscript.

(4) Line 185: Please revise “SD” to “standard deviation”

Results: Well-described

In Table 2, “group” à “Group”

In Table 3 and 4, “group” à “Group”, “baseline” à “Baseline”

In Table 5, “G” à “Group”, “baseline” à “Baseline”

Please revise all results to two decimal places from one or three decimal places in whole manuscript.

Line 258: Please remember that the main effect of time p=0.072 is not statistically significant.

Discussion

Please add several sentences of (1) application in this field and (2) more limitations of this study.

Conclusion

Line 321: (Reuben et al., 2004; Perera et al., 2005). à [50,51].

Comments on the Quality of English Language

I recommend that this manuscript should be edited by an English professional editor for more readable. There are some typo and grammatical errors.

Author Response

Reviewer 2

Dear authors,

This study was conducted to effects of chronic dietary protein manipulation on amino acids profile and position sense in elderly suffering type 2 diabetes mellitus. Fundamentally, this study is well described and use adequate methods and analyses. Moreover, it presents good data on this very important topic. I believe that this is very important issue in field of sports medicine and nutrition section. However, it needs more revisions. Some suggestions are recommended for the authors’ consideration.

Response

We would like to thank the reviewer for his/her supportive comments regarding our work and for the his/her comments that improved our manuscript.

Abstract

(1) Please you should add background of this study (1-2 sentences) in Introduction section.

Response

According to the reviewer’s suggestion, two sentences were added in the Abstract section. Now the relevant text reads:

“Dietary protein with adequate essential amino acids stimulates protein synthesis effectively and improves muscle mass. Musculoskeletal disorders in lower or upper limbs are not uncommon among patients with type II diabetes mellitus (T2DM).” (Abstract, lines 12-14).

(2) Line 22: “Position sense at the knee joint marginally improvement only in the intervention group by the 12th week (main effect of time p=0.072).” You have to revise “improvement” term because the p-value 0.072 is not statistically significant.

Response

The reviewer is right about the wording of this sentence. In the revised manuscript we have removed the words that imply significant difference:

“Position sense at the knee joint revealed a tendency for improvement in the intervention group by the 12th week (main effect of time p=0.072).” (Abstract, lines 24-26).

(3) Line 16: “1.0gr protein/kg/d and the intervention group received 1.2-1.5gr” à “1.0 gram protein/kg/day and the intervention group received 1.2-1.5 gram protein/kg/day.

Response

The relevant text was changed according to the reviewer’s suggestion. (Abstract, lines 18-19).

Introduction

(1) The authors should more explain the concept of chronic dietary protein manipulation on amino acids profile and position sense. Besides, the research gap between chronic dietary protein manipulation on amino acids profile and position sense in elderly and the authors should strengthen the gap illustration according to the prior research (added 3-4 paragraphs).

Response

According to the reviewer’s suggestion, the Introduction section was enriched in order to strengthen the research gap between the present investigation and the previous ones. The following sentences were added in the revised manuscript:

“Amino acids derived from proteins, in combination with glucose, stimulate insulin secretion leading to protein synthesis and the increase of muscle mass [10]. Note, however, that, unless amino-acids intake is balanced, adequate protein supplementation is not necessarily ingested effectively [11]. That is, not all amino acids will definitely contribute in protein synthesis.” (Introduction, par. 2).

“Musculoskeletal disorders in lower or upper limbs are not uncommon among patients with T2DM [21-23]. Inevitably, knee and shoulder joints have been repeatedly reported to be affected by T2DM resulting in deteriorated muscle strength and reduced mobility of the individuals [21-23].” (Introduction, par. 5, lines 98-101).

“Indeed, few recent studies have reported that T2DM patients exhibit impaired hip-joint or knee-joint position sense [31-32], and while sensorimotor training is effective to improve proprioception [33], no study has been found to take protein or any other diet into consideration.” (Introduction, par. 6, lines 116-119).

(2) If it is possible, please revise from “gr protein/kg/d” to “gram protein/kg/day” in whole manuscript.

Response

According to the reviewer’s suggestion, the “gr protein/kg/d” was revised to “gram protein/kg/day” throughout the manuscript.

Materials and methods: Well-written

(1) Line 88: Please provide exact IRB institute. It is not proper “local university ethics committee”.

Response

The exact IRB institute in now provided in the revised manuscript, that is, “National and Kapodistrian University of Athens”. (Materials and Methods, Participants description & eligibility criteria, line 195).

(2) In Table 1, Please revise “BMI” to “body mass index”.

Response

“BMI” was revised to “Body mass index” (Table 1)

(3) Line 119: Please revise “m2” to “m2” in whole manuscript.

Response

“m2” was revised to “m2” throughout manuscript.

(4) Line 185: Please revise “SD” to “standard deviation”

“SD” was revised to standard deviation” (Materials and Methods, statistical analysis, line 305).

Results: Well-described

In Table 2, “group” à “Group”. In Table 3 and 4, “group” à “Group”, “baseline” à “Baseline”. In Table 5, “G” à “Group”, “baseline” à “Baseline”

Response

All reviewer’s correction regarding Tables 2-5 were adopted in the revised manuscript.  

Please revise all results to two decimal places from one or three decimal places in whole manuscript.

Response

All numbers in the revised manuscript are now presented with two decimals, as suggested by the reviewer.  

Line 258: Please remember that the main effect of time p=0.072 is not statistically significant.

Response

The reviewer is right about the presentation of the result. In the revised manuscript we have removed the words that imply significant difference:

“Specifically, in the experimental group, there was a tendency for positive effect of time on knee joint position sense (p=0.072).” (Results, Position sense assessment, lines 906-907).

Discussion

Please add several sentences of (1) application in this field and (2) more limitations of this study.

Response

According to the reviewer’s suggestion the following sentences were added in the Discussion section of the revised manuscript:

“Overall, this study revealed that for patients suffering from T2DM a diet of protein with 1.2-1.5 gram/kg/day is more effective than current recommendations (0.8-1.0 gram/kg/day) to meet es-sential amino acids’ requirements and stimulate protein synthesis. The increase in protein synthe-sis explains the significant improvements in muscle strength for the group with the higher protein, in contrast to the group with the current recommendations.” (Discussion, par. 6, lines 1039-1043).

“Another limitation is that the recording of dietary recalls was relying on the memory and perception of each individual.” (Discussion, par. 6, lines 1045-1046).

Conclusion

Line 321: (Reuben et al., 2004; Perera et al., 2005). à [50,51].

Response

We would like to apologize for our mistake. In the revised manuscript only the number of the references are presented. (Discussion, par. 3, line 1020).

Round 2

Reviewer 2 Report

Comments and Suggestions for Authors

I believe that all of reviewers' issues are sufficiently addressed in the revised version. For this reason, I strongly would like to be recommended to accept the manuscript for publication.

Comments on the Quality of English Language

 Minor editing of English language required